# Red Light Resets the Expression Pattern, Phase, and Period of the Circadian Clock in Plants: A Computational Approach

**DOI:** 10.3390/biology11101479

**Published:** 2022-10-09

**Authors:** Ting Huang, Yao Shui, Yue Wu, Xilin Hou, Xiong You

**Affiliations:** 1College of Horticulture, Nanjing Agricultural University, Nanjing 210095, China; 2College of Sciences, Nanjing Agricultural University, Nanjing 210095, China

**Keywords:** plant circadian clock, red-light entrainment, photoperiod, differential equation model, phase shift

## Abstract

**Simple Summary:**

Progress in computational biology has provided a comprehensive understanding of the dynamics of the plant circadian clock. Previously proposed models of the plant circadian clock have intended to model its entrainment using white-light/dark cycles. However, these models have failed to take into account the effect of light quality on circadian rhythms, which has been experimentally observed. In this work, we developed a computational approach to characterizing the effects of light quality on plant circadian rhythms. The results demonstrated that red light can reset the expression patterns, phases, and periods of clock component genes. The circadian period, amplitude, and phase can be co-optimized for high-quality and efficient breeding.

**Abstract:**

Recent research in the fields of biochemistry and molecular biology has shown that different light qualities have extremely different effects on plant development, and optimizing light quality conditions can speed up plant growth. Clock-regulated red-light signaling, can enhance hypocotyl elongation, and increase seedling height and flower and fruit productivity. In order to investigate the effect of red light on circadian clocks in plants, a novel computational model was established. The expression profiles of the circadian element *CCA1* from previous related studies were used to fit the model. The simulation results were validated by the expression patterns of *CCA1* in *Arabidopsis*, including wild types and mutants, and by the phase shifts of *CCA1* after red-light pulse. The model was used to further explore the complex responses to various photoperiods, such as the natural white-light/dark cycles, red/white/dark cycles, and extreme 24 h photoperiods. These results demonstrated that red light can reset the expression pattern, period, and phase of the circadian clock. Finally, we identified the dependence of phase shifts on the length of red-light pulse and the minimum red-light pulse length required for producing an observable phase shift. This work provides a promising computational approach to investigating the response of the circadian clock to other light qualities.

## 1. Introduction

Plants have evolved a sophisticated timing mechanism, known as the circadian clock, to sense, respond to and tune to the switching between day and night due to the rotation of the earth [1,2]. This mechanism allows plants to anticipate and coordinate their biological processes with diurnal rhythms. In *Arabidopsis thaliana*, the expression of nearly one-third of the genome is under circadian regulation [3,4,5]. Variation in the circadian period, phase, and amplitude variations may affect the expression and stability of several circadian related outputs, such as flowering time [6]. Co-optimizing the circadian period, amplitude, and phase can help with breeding and engineering. Increasing the circadian period, and therefore, delaying the phase, confers benefits at higher latitudes for cultivation [7].

Conceptually, the circadian clock system consists of light-signal receptors, a core oscillator that maintains a roughly 24 h rhythm even in the absence of input signals, and physiological/biochemical outputs [8]. Among the various input signals, light is critical and can reset and entrain the plant circadian clock [9]. Light regulates plant circadian clock components at transcription, translation, and post-translational levels [10].

Recent studies have shown that different light qualities have extremely different effects on plant development [11]. Although plants on normal days seldom encounter light of a single wavelength, monochromatic light exposure has been applied for efficient breeding of high quality in modern plant facilities for artificial cultivation [12,13,14,15]. Breeders use supplementary lighting by fluorescent lamps and LEDs to improving light conditions. Red light serves as one of the key light-quality factors, which affects plant growth, development, product quality, and flavor [16,17]. It has been reported that pre-sowing the red-light stimulation of seeds could improve germination rate [18]. Supplementary red light leads to the earlier ripening of tomato fruit [19]. Nitrate is essential for normal plant growth but harmful for human health. The nitrate concentration in plants can be effectively reduced under red-light exposure to achieve food safety [20]. At the same time, plant growth and safety can be balanced under red illumination [21]. Irradiating table grapes with red light contributes to their extended postharvest shelf-life and improves the commercial quality of the grapes [17]. The affinity between red light and clock responses in plants can enable various physiological output processes in a timely manner to deal with light changes, including periodic or transient stimuli [22].

In the past two decades, a vast number of experiments have been conducted to reveal the molecular mechanism of the plant circadian clock response to light inputs [23,24,25,26,27,28]. Although a tremendous number of experimental facts have been accumulated, large numbers of interactions and regulations make it difficult to see the full picture of plant circadian clocks. As a recipe, mathematical models provide a convenient and efficient approach to exploring the dynamic characteristics of the network, where every clock component can be analyzed simultaneously in a quantitative and qualitative way. Models can be constructed for the circadian oscillator at the molecular, cellular, or tissular level. The pioneering work by Locke et al. [29] proposed a simple model describing the *Arabidopsis* central oscillator based on reciprocal regulation between CIRCADIAN CLOCK-ASSOCIATED 1 (CCA1)/LATE ELONGATED HYPOCOTYL (LHY) and TIMING OF CAB EXPRESSION 1 (TOC1). As new clock components and their regulations were identified experimentally, many improved, refined, and simplified models were developed in sequence [9,10,30,31,32]. Among the notable improvements was the compact model by De Caluwé et al. [10], where functionally similar clock genes were grouped and merged into a single component according to the similar nature of their behaviors.

These models are all designed for white-light signals. However, recent studies have found that different light qualities have quite different effects on plant development, and optimizing light quality conditions can speed up plant growth. Red light exerts peculiar impact on plants by resetting and entraining the circadian clock [33,34,35]. In this paper, we developed a computational model for the specific effect of red light on the plant circadian clock. Using the model, we predicted the expression profiles of the core circadian gene *CCA1* in constant red light (RR) and the phase shifts caused by red-light pulses. These predictions were confirmed with experimental data on *Arabidopsis*. Additionally, we validated that the red-light induced expression of *CCA1* was reduced in TOC1-OX and *toc1* mutant lines. To further explore the complex dynamics under different red-light/dark cycles, we compared the dynamics of the core circadian clock element *CCA1* under respective RR, constant white-light (LL), and constant dark (DD) conditions, both for the wild type (WT) and mutant. We then continued to simulate the multiple circadian behaviors under respective red-light/dark (RD), red/white/dark (RLD), or white/red/dark (LRD) and non-24 h cycles, capturing the periods and phases. Finally, phase-response curves (PRCs) were drawn to identify the phase shifts produced by various lengths of red-light pulses at indicated times.

## 2. Materials and Methods

### 2.1. The Molecular Mechanism of Red-Light-Entrained Plant Circadian Clock

The prototype of our model is the compact model of the plant circadian system (PCS) for the white-light-entrained plant circadian clock proposed by De Caluwé et al., hereafter referred to as the De Caluwé model [10], which includes four pairs of clock genes: *CL* (*CCA1* and *LHY*), *P97* (*PSEUDO-RESPONSE REGULATOR 9* (*PRR9*) and *PRR7*), *P51* (*PRR5* and *TOC1*), and *EL* (EARLY FLOWRING 4 (*ELF4*) and *LUX ARRYTHMO* (*LUX*)). Two MYB-like domain transcription factors, CCA1 and LHY, are expressed at dawn. They directly inhibit the expression of early (peak expression in the morning) and late (peak expression in the evening) clock genes, as well as self-expression [36]. The PRR family members (PRR9, PRR7, PRR5, and TOC1) are expressed in order, with mutual inhibition and suppression of the expression of *CCA1* and *LHY* [37]. At night, TOC1 inhibits the above-mentioned components, appending LUX and ELF4 [38]. We modified the De Caluwé model by switching the activation of CL in *P97* to inhibition and attaching the self-inhibition of *CL* [35].

It was reported that ELF3 is involved in integrating red-light input into the clock, e.g., overexpressing ELF3 weakens the sensitivity of the clock entrained to red-light-mediated resetting cues [4]. On one hand, CCA1 represses the expression *ELF3* in the morning by directly binding to its promoter [39]. On the other hand, *ELF3* and *ELF4*, whose expressions are necessary for the red-light-induced expression of both CCA1 and LHY, are in turn regulated by light signaling [40,41]. Red light controls the expression of clock elements primarily through the major red-light photoreceptor phytochrome B (phyB). For instance, the overexpression of phyB stabilized ELF3 proteins [42], and Green Fluorescent Protein (GFP) results revealed that phyB physically interacted with the plant clock elements CCA1, LHY, TOC1, LUX, and ELF3 [24].

Therefore, for the purpose of modeling red-light entrainment, it is necessary to complemented the *ELF3* gene to the De Caluwé model. The resulting circadian clock scheme is shown in Figure 1.

### 2.2. Formulation of Model Equations

Based on the schematic network in Figure 1, we set out to mathematically model the dynamics of the red-light-entrained circadian clock. Instead of white-light input in the De Caluwé model [10], we replenished red-light cues to the main clock. According to experiments, red light regulates circadian clock components by affecting the rates of transcription, translation, and post-translation modification [43,44,45,46]. Protein degradation is modeled based on the fact that phyB regulates the circadian clock components through direct protein–protein interactions.

The new computational model consists of 11 ordinary differential equations (ODEs) with 57 parameters (see Appendix A). Appendix A depict the temporal expression of the mRNA and protein of the clock elements: CL (CCA1 and LHY), P97 (PRR9 and PRR7), P51 (PRR5 and TOC1), EL (ELF4 and LUX), and ELF3. The Appendix A describes the activation percentage of the light-sensitive protein P, simplified from the equation introduced by Locke et al. [29]. In modeling gene regulation, the transcription reaction follows nonlinear Hill dynamics, while the translation and degradation reactions follow the linear mass-action law. In addition, the effect of the two inhibitors A and B on C (e.g., the two inhibitors P97 and P51 on *CL*, see Appendix A) is mutually exclusive in Hill-type terms. The light cues are represented by the variables *L* (white light), *D* (darkness), and *R* (red light). They take values either 1 or 0 according to the light condition. To be specific, *L* = 1, *R* = 0, *D* = 0 when the white light is present; *L* = 0, *R* = 1, *D* = 0 when the red light is present; and *L* = 0, *R* = 0, *D* = 1 when it is in dark. A total of 35 parameters were adopted from the De Caluwé model [10]. The transcript levels of *CCA1* under a skeleton photoperiod (Appendix A) were used to fit the other 22 red-light associated parameters, which were estimated by minimizing the cost function, as described in Appendix A.

### 2.3. Database and Simulation Tools

The expression profiles of the circadian element *CCA1* under constant red-light exposure and the skeleton photoperiod were extracted from the article of Nimmo et al. [47]. The phase shift values of the circadian element *CCA1* under red-light pulses were extracted from Ohara et al. [48] using the software ImageJ (https://imagej.nih.gov/ij/ (accessed on 10 September 2020), see Appendix A for detailed steps). In the experiments, plants were illuminated with white/red light provided by LEDs (Luxeon Star 460 nm and 627 nm, respectively) at a fluence rate of 100 and 20/40 μmol m^−2^ s^−1^, respectively. The seedlings were subjectively entrained under 12 h white-light/12 h dark cycles for 5 days before being released to different light/dark photoperiods.

We simulated our ODE model utilizing Python 3.10.4, a Java-based software that is freely available for the researchers (https://www.python.org/downloads/ (accessed on 1 March 2020). The 4th-order Runge–Kutta method was used to obtain the numerical solutions of the ODEs (See Appendix A).

## 3. Results

### 3.1. Model Predictions and Validation

#### 3.1.1. Rhythmic Expression Patterns

The transcript levels of *CCA1* under a skeleton photoperiod (Appendix A) were used to fit our model. With our model, we then simulated the rhythmic expression patterns of the clock component *CCA1* under constant red light for the wild type and *elf4* mutant.

According to the experimental treatment, the WT lines were entrained in 12 h white-light/12 h dark cycles for 5 days, and then, transferred to constant red light at ZT = 0 (Figure 2A) or at ZT = 12 h (Figure 2B). In both cases, the simulated curves were able to retain sustained rhythmicity, consistent with the experimental data.

The evening complex has been reported to integrate light signals to regulate the clock [34]. To verify that the clock of the *elf4* mutant shifted from 12 h white-light/12 h dark cycles to constant red light, we simulated the expression of *CCA1* with the same light input mode as WT (Figure 2A). The simulated results demonstrated that the *elf4* mutant displayed sustained oscillation with a damped amplitude (Figure 2C).

#### 3.1.2. Phase Shifts Caused by Red-Light Pulses

Phase, as a main characteristic of oscillators, is associated with daily variations in red-light pulse [48]. First, we examined the phase shifts of our model by simulating the phase-response curves (PRCs) (the calculation method can be found in Appendix A). The stimulus conditions included a “red-pulse” with the red light switched on for 1 h in DD (Figure 3A) and a “dark-pulse” with the red light switched off for 2 h in RR (Figure 3B). The model had been entrained under a light/dark cycle (12 h white light/12 h dark) for 120 h before transferring to the DD or RR condition. The simulated PRC reproduced type-0 (Figure 3A) and type-1 (Figure 3B) patterns for the red-light pulse and dark pulse, respectively, capturing the essential feature of the experimental PRC.

#### 3.1.3. Red-Light Induction of CCA1 Expression with TOC1/ELF4 Overexpression and Mutation Lines

*CCA1* gene expression has been found to be induced by light signal and may have different profiles in different *Arabidopsis* lines [40,49]. Moreover, TOC1 and ELF4 have been reported to act negatively and positively, respectively, on *CCA1*/*LHY* expression under red-light exposure [40,46]. Here, we examined and predicted the effect of a 1 h red-light pulse on *CL* transcriptional levels in WT, TOC1-OX, the *toc1* mutant, ELF4-OX, and the *elf4* mutant. As shown in Figure 4, the *CCA1* mRNA levels in WT were induced strongly after a 1 h red-light pulse. The transcriptional levels of *CCA1* dropped rapidly after being released back to the dark for 1 h. A slight increase in CCA1 induction was observed in both TOC1-OX and the *toc1* mutant (Figure 4A), with a clear reduction compared with WT. This result was consistent with previous work [40]. By contrast, the extents of *CCA1* mRNA were predicted to be enhanced in ELF4-OX and the *elf4* mutant compared with WT (Figure 4B). These results indicate that except for TOC1, ELF4 is always involved in the red-light-mediated, phytochrome-dependent regulation of *CCA1*/*LHY* expression.

### 3.2. Red-Light Reset of the Expression Pattern, Period, and Phase of CL and P51 under Different Photoperiods

#### 3.2.1. Clock Reset under Red-Light Free-Running Period in WT

In the free-running simulations under constant conditions, our model displayed self-sustained circadian rhythms under RR conditions (Figure 5A), and damped oscillations under either LL (Figure 5B) or DD (Figure 5C) condition. This suggested that continuous red-light stimulation would reset the expression pattern of the circadian clock. We next attempted to recapitulate in our model the differences in clock periods and phases. The Fourier fitting gave the theoretical free-running periods 26.4 h under RR, 24.41 h under LL, and 26 h under DD. In addition, the simulated phases of CL and P51 mRNA were 2.5 h and 18.27 h under RR, 2.95 h and 15.18 h under LL, and 2.55 h and 16.66 h under DD, respectively. In addition, compared with the other two light treatments, the steady-state value of CL mRNA under LL was at the lowest level. These results showed that constant red light could reset the period and phase of the circadian clock.

#### 3.2.2. Clock Reset under Red-Light/Dark Cycles in WT

Except for the dynamics of the clocks under free-running conditions, it is more important to identify the characteristics of clocks under varying light conditions similar to those in the natural world. Therefore, we simulated the time courses of clock component *CCA1* under a short (Figure 6A), medium (Figure 6B), and long day (Figure 6C). An investigating alternative was to replace the white-light/dark cycles with the red-light/dark cycles. We then tried to describe the similarities and differences in the waveform, phase, and period. The shapes of the expression curves were obviously different. The temporal curves in white-light/dark cycles were smooth (black lines), whereas small shoulders and minor spikes appeared beside the peaks in red-light/dark cycles (red lines). The phases and periods under the distinct red-light/dark cycles are listed in Appendix A, suggesting that the period and phase of the circadian clock could be changed via light-quality variation.

#### 3.2.3. Clock Reset under Red-Light/Dark Cycles in Single or Double Mutant

In order to estimate the defects associated with a loss-of-function mutation of the main clock genes under red-light/dark cycles, we observed those features in single and double mutants differently to those under white-light/dark cycles. In the De Caluwé model, there are three single mutants with the same or very similar defects, and a double mutant with qualitatively similar but more pronounced version of the same phenotype. The single mutants were simulated by dividing the relevant mRNA synthesis rate by 2 (Figure 7B–D). For simulated double mutants, the synthesis rate was multiplied by 0.1 (Figure 7A).

Under white-light/dark cycles, the PRR5/TOC1 transcript had a phase advance in the *cca1 LHY* double mutant compared with that in WT, whereas no phase shift occurred under red-light/dark cycles. Moreover, the inhibition of CCA1 and LHY in evening-phased genes weakened or disappeared obviously under red light cycles, leading to the peak time of PRR5/TOC1 occurring at dawn (Figure 7A). Experimentally in white-light/dark cycles, the *prr9* and *prr7*, *prr5* and *toc1,* and *elf4*/*lux* single mutant had slight defects, both in clock gene expression levels and in periods (25 to 26 h) [50,51,52]. Inversely, the waveform, phase, and transcriptional level exhibited a great difference between the single mutant and WT when released to red-light cycles (Figure 7B–D). Phase advance (Figure 7C) and delay (Figure 7D) were observed in red-light entrainment. Similarly, red light might be more helpful for the mutant to attenuate the repression of evening-phased genes on CCA1/LHY.

#### 3.2.4. Clock Reset under Extreme 24 h Red-Light Photoperiods

Entrainment to different photoperiods can affect the dynamics of the clock. The frequency will demultiplicate when the clock was subjected to a short cycle that is outside of its range of entrainment [10]. In this short cycle, the clock would be entrained to the short period. The model could generate this behavior (Appendix A). In both the wild type and *elf3* mutant plant clock oscillated with a 12 h period following a switch from standard 24 h (12 h white light/12 h dark) conditions to 12 h (6 h white light/6 h dark) in keeping with the experimental data. Additionally, this result overturned the previous simulation, which reported that the frequency under the short period was close to the integer multiple of the free-running period [53]. The *elf3* mutation had little effect on the relative abundance of CCA1 mRNA compared with that in the wild type.

Lightly affected/rhythmic *elf3* mutant was immediately entrained to the short period under the white-/red-light cycle, in the same way as that in the wild-type (Appendix A). Compared with its expression in the white-light/dark short cycle, the *CCA1* transcript was maintained at a lower level, especially in the *elf3* mutant line (Appendix A). This result was owing to *ELF3* expression, which was essential for *CCA1*/*LHY* to generate highly abundant oscillation under red light [41].

Using our model, we can simulate the clock dynamics under different red-light photoperiods on the heels of exposure to a 24 h cycle. The temporal evolutions of *CL* (*CCA1*/*LHY*) (Figure 8A) and *P51* (*PRR5*/*TOC1*) (Figure 8B) mRNA levels were simulated during a 12 h light/12 h dark cycle followed by red-light photoperiods ranging from 3 to 21 h. The length of the red-light period had a significant effect on the expression waveform of clock components and their expression phases. We then calculated the shift differences between the anterior phase and posterior phase (Figure 8C,D; Appendix A) on day 5 of entrainment. The phase changes in *CCA1* mRNA increased linearly in the red-light length. Nevertheless, but with a jump on the long red-light day. The phase of the first peak of TOC1 expression after release into short red-light entrainment was largely independent of the preceding entraining conditions, and was different from the downward parabolic type on the long red-light day. On the other hand, the dynamics under constant red light after exposure to a 24 h cycle, regardless of the length of entraining light period, was simulated (Appendix A). The white-light/dark cycle in the photoperiods ranged from 3 to 21 h, followed by release into continuous red light. It was seen that the length of white light had little effect on the expression waveform of clock components, but produced phase shifts. This result was consistence with the white-light free-running pattern [54].

#### 3.2.5. The Expression Pattern of CCA1 Determined by the Orders of Red-Light Input

In order to investigate the influence of different light modes and light quality on clock expression, we selected red- and white-light inputs with different light lengths and orders for numerical simulation. The simulation results showed that the input order of red light and white light affected the expression pattern of the clock. The order of red/white light pulses and light pulse lengths (ranging from 3 h to 9 h) did not affect the period of the clock (Appendix A). From the simulated waveforms, the expression of the clock component CCA1 transited smoothly from white light to red light (Figure 9B), while there were several spikes as red-light input preceded white light (Figure 9A).

#### 3.2.6. Phase Shifts Controlled by the Moments and Lengths of Red-Light Stimulation

In order to further explore dependence of phase shift on the stimulus time and the length of the red pulse on the phase shift in the white-light/dark cycle, the system was stimulated at the indicated times with red-light pulses of 0.25, 0.5, 0.75, 1, 1.25, 1.5, 1.75, 2, 2.25, 2.5, 2.75, 3, 3.25, 3.5, 3.75, and 4 h (Figure 10A). The observed phase shifts were characterized as type-1 resetting, consistent with experimental observations. Phase delays occurred around the time when darkness switched to white light, and phase advances took place around the time when white light turned to darkness. Simultaneously, we calculated the minimum pulse length of the phase shift at diverse stimulus times. The simulation results revealed that red-light stimulation of different durations (within 4 h) in ZT1-ZT5 would not alter the phase (Figure 10B). The longest duration of red-light stimulus to produce an explicit phase shift was at ZT6. If the red pulse duration was 0.25 h, the phase of the clock component remained unchanged regardless of the stimulus time.

## 4. Discussion

In this study, we present a computational model to characterize the effect of red light on the plant circadian clock. This model is capable of reproducing key characteristics of the *Arabidopsis thaliana* circadian clock, including wild-type and mutant phenotypes under constant red-light and white-light/dark skeleton photoperiods. The model can be applied to the circadian clock under diverse artificial red-light environments, such as red-light/dark cycles, red-light/white-light/dark cycles and red-light pulses [33]. In our model, the De Caluwé model as a whole and the additional *ELF3* gene forms a new feedback loop, which requires 2 additional equations to describe. This extension structure allows us to capture more complex response behavior of the circadian clock for various red-light inputs (Figure 2 and Appendix A), whereas the De Caluwé model is only suit for white light photoperiods

In the negative-feedback loop among CCA1/LHY, PRR9/PRR7, and PRR5/TOC1, the expression of *CCA1*/*LHY* mRNA was supposed to be activated and inhibited in the *toc1* mutant and TOC1-OX lines. However, a clear reduction in the extent of *CCA1* induction was observed both in the mutant and overexpression lines after 1 h of red-light pulse (Figure 4A), suggesting that PRR9/PRR7 might be able to enhance the suppression of *CCA1* depending on the changes in red-light quality. This is consistent with the proposal that PRR9/PRR7 appears to be involved in red-light quality-dependent circadian entrainment [55]. On the contrary, the *elf4* mutant displayed higher activation of *CCA1* in response to the red-light pulse (Figure 4B). There might be an underlying clock defect that affects the gating of this red-light response pathway [52]. We speculate that the inhibition of *CCA1* abundance by PRR9/PRR7 would be reduced in the *elf4* mutant.

Under the constant red-light, the plant circadian clock retains a robust periodic oscillation, in contract to a damped oscillation under constant white light or constant dark. We proved the reliability of the numerical simulations through mathematical analysis. The root of the nonlinear equation system was first solved for the steady-state, and the eigenvalues of the Jacobian matrix at steady-state through eigenvalue analysis, were found (See Appendix A). Since there are eigenvalues with positive real parts, it means that the steady-state is unstable. At the same time, a pair of conjugate complex roots indicates that the system can retain robust oscillation.

An important task is to determine the conditions to assure sustained oscillations in clock gene expression under constant red light. Hopf bifurcation (Appendix A) and parameter sensitivity analysis (Appendix A) were carried out to identify the dependence of dynamics of the clock on parameters. It suggested that oscillations occur for suitable range of the EL protein degradation rate $d_{4R}$ [56]. The CL mRNA of system moved first from the stable steady state with $d_{4R}$ less than 0.1 to the unstable steady state in the interval from 0.1 to 1.2, and then to the stable steady state again as $d_{4R}$ increased (Appendix A). The P51 mRNA level presented as a function of parameter $d_{4R}$, oscillating in the umbrella-shaped area (Appendix A), as well. And the corresponding periods decreased as $d_{4R}$ increased and eventually flatten out in less than 20 h (Appendix A). The time evolutions for $d_{4R}$ in different domains were shown in Appendix A.

Phase sensitivity analysis (Appendix A) results demonstrated that the degradation rates of CL mRNA ($k_{1R}$), CL protein ($d_{1R}$), P97 protein ($d_{2R}$), and EL protein ($d_{4R}$) were sensitive to phase under RR. Period sensitivity analysis (Appendix A) results showed that the ELF3-induced synthesis rate of CL mRNA ($v_{1B}$); red-light-induced synthesis rate of ELF3 mRNA ($v_{5R}$); degradation rates of CL mRNA ($k_{1R}$), ELF3 mRNA ($k_{5}$), CL protein ($d_{1R}$), EL protein ($d_{4R}$) and ELF3 protein ($d_{5R}$); and translation rate of ELF3 ($p_{5}$) were sensitive to period under RR. Based on the sensitivity regions of the parameter values, we can extend the red-light acclimation model of *Arabidopsis* to different varieties of the same plant or even other plants. Red light would affect the variations in protein degradation rates in a proteasome-dependent pathway [57]. The dynamical behaviors of different red-light-acclimated plants can be recognized through the fluctuation of degradation rate constants of proteins within the sensitivity regions (Appendix A).

To further study the responses, the role of multiple light inputs in seasonal adaptation could be explored via a systematic analysis of entrainment in different photoperiods [58] and other monochromatic light entrainments, such as blue light [59]. Red and blue light are the two most effective waves for plant photosynthesis [60]. Blue light has different effects on the circadian clock. For example, the period of the clock that is much less sensitive to blue light than to red light [61]. Additionally, the phases and expression patterns of core oscillators under blue-light pulses were different from those under red-light pulses [47,48]. The robustness of the circadian oscillations in the presence of both molecular and environmental variability can readily be investigated through stochastic simulations [62,63].

This study may contribute to the understanding of circadian entrainment in a natural environment. Theoretical studies show that the color of sunlight changes throughout the day. For example, during the day, sunlight contains more blue than red light. At sunset (and sunrise), sunlight has more red light than blue light. According to the actual measurement of the sunlight and sky-light spectrum in an urban area, the peak value of the solar spectrum changes regularly in a day under general atmospheric conditions, from 666 nm at sunrise to 530 nm during the daytime, and then, to 642 nm at sunset [44]. Based on the difference between the spectrum and the local sunrise and sunset times (sunrise, 6:00–8:00; daytime, 8:00–16:50; sunset, 16:50–17:32), the natural sunlight input is regarded as a red light/white light/red light/dark cycle.

To recapitulate the percentage of the wavelengths in a diurnal cycle, we added 2 h of red-light stimulation at dawn and 1 h red-light stimulation at dusk in 12 h white light/12 h dark cycles (Appendix A). This gave no effect on the expression mode and phase of *CL* (*CCA1*/*LHY*), but an explicit delay in the phase of *P51* (*PRR5*/*TOC1*) transcript. Moreover, despite the normal peak in the beginning of night, an additional peak of *P51* occurred at the end of sunrise red light, which verified the observation that *P51* may have several expression-peak phases under a subjective natural light cycle [64]. The model can be applied to multiple-light entrainment in various regions with different latitudes and longitudes.

For the sake of simplicity, our model assumes that the photosensitive protein phyB only responds to red light. However, circadian clocks are sensitive to different light qualities, such as far red light and blue light [65]. On the other hand, light intensity is a critical factor in the behaviors of circadian oscillators [5]. For example, in a clock element mutant in various intensities of red light may lead to arrhythmia, whereas blue light may induce a faster clock [49]. In addition, circadian clock feedback to the red-light-photosensitive protein phyB needs to be incorporated into the model in an appropriate way based on experimental facts [66]. Based on the red-light-entrained model with precise data on diurnal changes in the spectrum, we may further understand how plants work in the natural environment.

## 5. Conclusions

Genes subjected to circadian clock regulation can be entrained under various modes of light input. Genes involved in the core oscillator respond to red-light changes by adjusting their expression patterns, periods, and phases. Currently available models of the plant circadian clock are limited, especially in red-light input, and our red-light-entrained model is an important supplement. This paper mainly aimed to construct a schematic structure of the circadian clock response to red-light signals, seek a set of optimal parameters, conduct model-reliability analysis, and introduce the red-light-sensitive protein to the circadian clock. Then, the expression profiles of the core circadian element *CCA1* and phase shifts under red-light pulses were validated with experimental data from previous studies. Using the model, dynamics, and phase properties of the circadian clock under various photoperiods were further explored, suggesting that expression pattern, period, and phase of the circadian clock could be reset. Some predicted phenomena need to be confirmed in the subsequent experiment of the future research.

## Figures and Tables

**Figure 1 biology-11-01479-f001:**
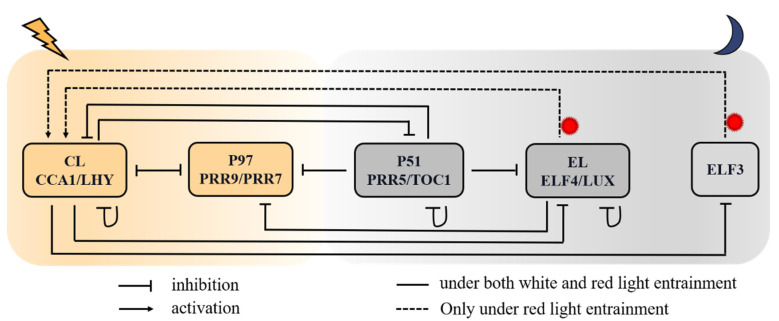
Schematic red-light-responsive circadian network in *Arabidopsis*. The genes in one box have similar expression profiles, regulators, targets, and defects in loss-of-function mutant lines. Solid lines with blunt ends indicate that genes function as repressors in the negative-feedback loops. Dotted lines and arrows indicate genes acting as activators in the regulatory network. The lightning and moon symbols represent light and darkness, respectively. *CL* (*CCA1*/*LHY*) and *P97* (*PRR9*/*PRR7*) are morning-peaking genes, and the rest are afternoon-peaking or night-peaking genes. The dotted lines indicate regulatory relationships only under red light.

**Figure 2 biology-11-01479-f002:**
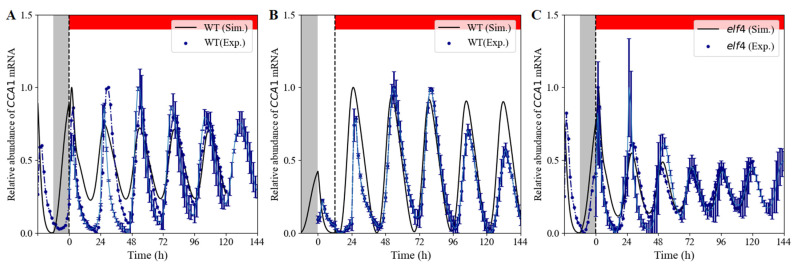
The expression profiles of the core circadian element *CCA1* under constant red light (RR). Wild-type (WT) and mutant lines were entrained for 120 h under white-light/dark cycles. (**A**) The WT lines were transferred into RR at ZT0. (**B**) The WT lines were transferred into RR at ZT12. (**C**) The *elf4* mutant lines were transferred into RR at ZT0. The basic parameters are shown in Appendix A. In the *elf4* mutant simulation, the maximum synthesis rate in white light ($v_{4L}$) and red light ($v_{4R}$) of *EL* mRNA are respective 0.294 and 2.63 versus 1.47 and 6.1 in WT. The grey and red bars represent darkness and red-light treatment, respectively. The black solid lines and the blue circles indicate simulated expression (Sim.) and experimental data (Exp.), respectively. Transcript levels are means ± SD for *n* = 8~26 clusters of *Arabidopsis* plants in 3~6 biological replicates.

**Figure 3 biology-11-01479-f003:**
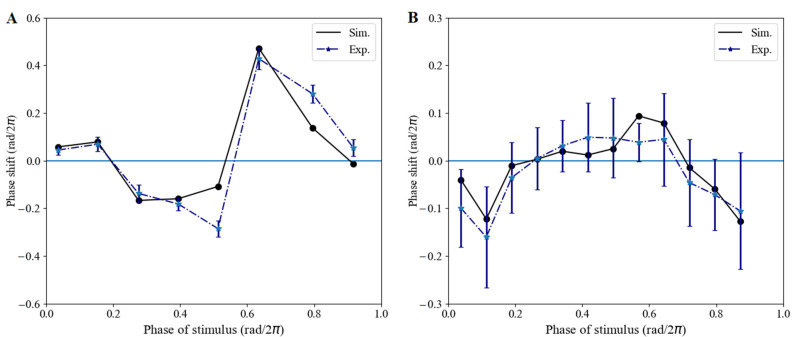
Phase shift validation. (**A**) WT lines were first entrained in continuous darkness; then, a red-light pulse was imposed for 1 h at an indicated time within 24 h. (**B**) WT lines were first exposed to constant red light with a 2 h dark shock at an indicated time. Black dots denote the simulated PRC (Sim.), and blue triangle and dotted line denote the experimental PRC (Exp.). Phase shift values are means ± SD in three biological replicates.

**Figure 4 biology-11-01479-f004:**
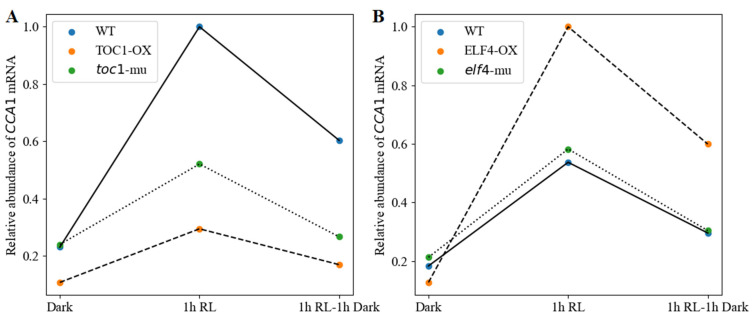
The red-light-mediated induction of *CCA1* in WT, TOC1-OX, *toc1*-mu, ELF4-OX, and *elf4*-mu. (**A**) The simulated expression of *CL* mRNA in WT, TOC1-OX, and *toc1*-mu. (**B**) The simulated expression of *CL* mRNA in WT, ELF4-OX, and *elf4*-mu. In the experiments, plants maintained for 5 days in the dark were treated for 1 h with red-light stimulation (RL; 40 μmol m^−2^ s^−1^) followed by 1 h dark exposure. Transcript levels were simulated in WT with the following basic parameters: TOC1-OX (the synthesis rate $v_{3}$ was doubled), *toc1*-mu (the synthesis rate $v_{3}$ was halved), ELF4-OX (the synthesis rates $v_{4L}$ and $v_{4R}$ were doubled), and *elf4*-mu (the synthesis rates $v_{4L}$ and $v_{4R}$ were halved). WT, TOC1-OX, *toc1*-mu, ELF4-OX, and *elf4*-mu indicate the wild type, the overexpression of TOC1, the *toc1* mutant, the overexpression of ELF4, and the *elf4* mutant, respectively.

**Figure 5 biology-11-01479-f005:**
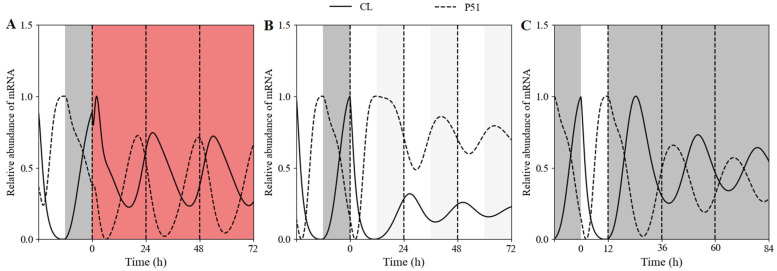
The temporal evolution of circadian clock in free-running period. The WT lines were entrained in white-light/dark cycles (12 h light/12 h dark) for 5 d, and then, transferred to 24 h constant light on day 5. (**A**) The WT lines were transferred into RR at ZT0. (**B**) The WT lines were transferred into LL at ZT0. (**C**) The WT lines were transferred into DD at ZT12. The grey, red, and white bands represent subjective continuous darkness (DD), constant red light (RR), and constant white light (LL), respectively. The solid and dashed lines represent the simulated transcript levels of *CL* (*CCA1*/*LHY*) and *P51* (*PRR5*/*TOC1*) under different free-running conditions. All values are normalized to their respective maxima.

**Figure 6 biology-11-01479-f006:**
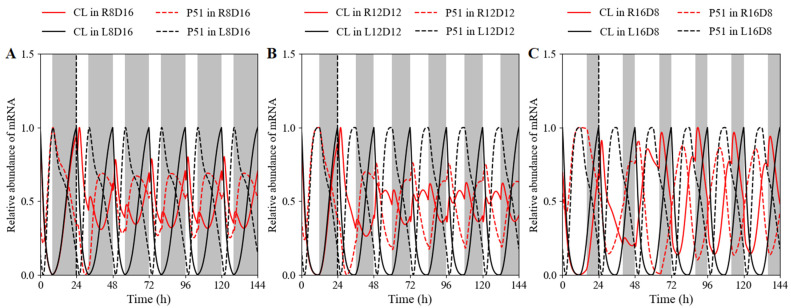
Different dynamic behaviors of the clock elements *CL* and *P51* under red-light cycles in comparison with those under white-light cycles. The simulated expression profiles of *CL* (*CCA1*/*LHY*) and *P51* (*PRR5*/*TOC1*) mRNA on (**A**) short days (8 h light/16 h dark), (**B**) intermediate-light-length days (12 h light/12 h dark), and (**C**) long days (16 h light/8 h dark). R, L, and D indicate red light, white light, and dark, respectively. The solid and dashed lines represent the simulated transcript levels of *CL* (*CCA1*/*LHY*) and *P51* (*PRR5*/*TOC1*), respectively. All values were normalized to their respective maxima.

**Figure 7 biology-11-01479-f007:**
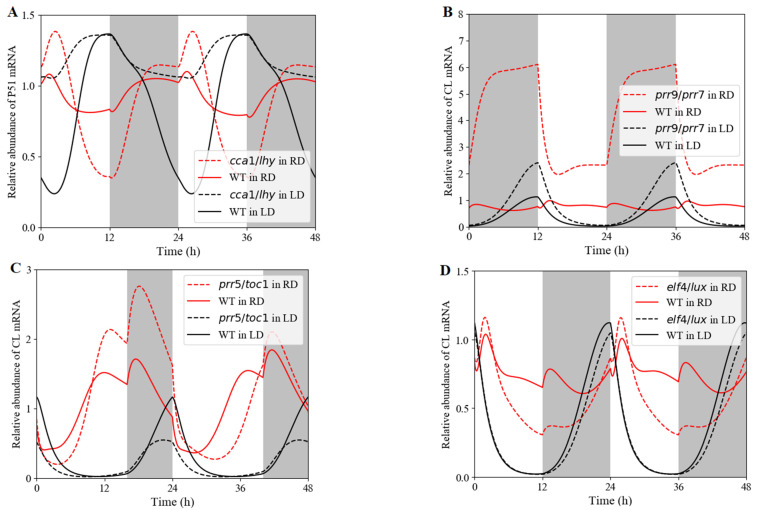
Expression of clock genes in loss-of-function mutants under either white-light/dark or red-light/dark cycles. Simulated mRNA levels in various clock mutants (dashed lines) and their respective wild types (solid lines) under red-light cycles (RD, red lines) and white-light cycles (LD, black lines). (**A**) Simulated *PRR5*/*TOC1* expression in the *cca1*/*LHY* double mutant. (**B**) *CCA1*/*LHY* expression in the *prr9*/*prr7* single mutant. (**C**) Simulated *CCA1*/*LHY* expression in the *prr5*/*toc1* single mutant. (**D**) Simulated *CCA1*/*LHY* expression in the *elf4*/*lux* single mutant. The grey and white bars represent darkness and light (white light (L) or red light (R)) treatments, respectively.

**Figure 8 biology-11-01479-f008:**
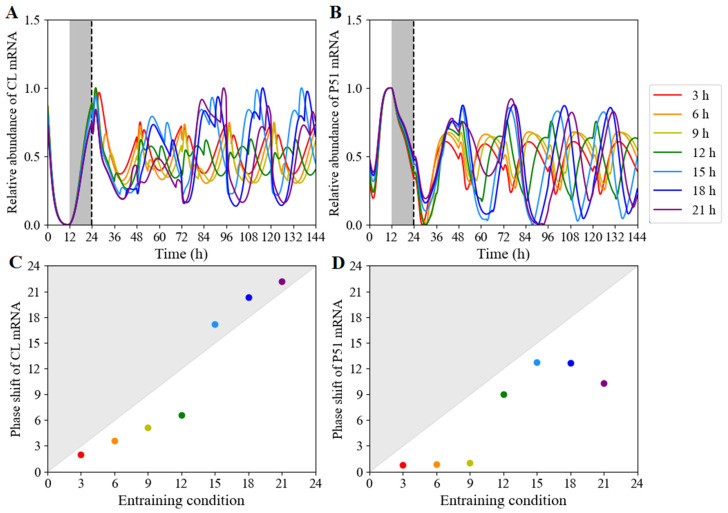
*CCA1*/*LHY* and *PRR5*/*TOC1* expression profiles and phase shifts under different red-light/dark photoperiods. (**A**,**B**) The time courses of respective *CL* (*CCA1*/*LHY*) and *P51* (*PRR5*/*TOC1*) transcription levels under red-light photoperiods ranging from 3 to 21 h at intervals of 3 h. The system had been entrained to 12 h white-light/12 h dark cycles for 5 days, transferring into red light. (**C**,**D**) The corresponding phase shifts of *CL* (CCA1/LHY) and *P51* (PRR5/TOC1), respectively. The grey and white bars represent darkness and red-light treatment, respectively.

**Figure 9 biology-11-01479-f009:**
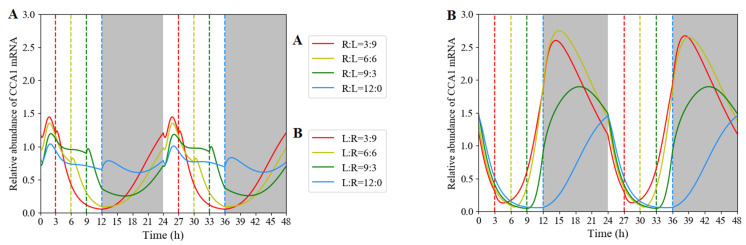
Simulated expression of *CCA1* transcript under various red/white and white/red cycles in 12 h light interval with fixed 12 h darkness. The advanced red-light input (**A**) and advanced white-light input (**B**) were entrained to different red-light/white-light/dark photoperiods, where red light ranged from 3 to 12 h in increments of 3 h. R:L = 3:9 (red vertical dashed lines in panel (**A**) and L:R = 3:9 (red vertical dashed lines in panel (**B**) represent 3 h red light/9 h white light/12 h dark and 3 h white light/9 h red light/12 h dark in a diurnal cycle. R:L = 6:6 (yellow vertical dashed lines in panel (**A**) and L:R = 6:6 (yellow vertical dashed lines in panel (**B**) represent 6 h red light/6 h white light/12 h dark and 6 h white light/6 h red light/12 h dark in a diurnal cycle. R:L = 9:3 (green vertical dashed lines in panel (**A**) and L:R = 9:3 (green vertical dashed lines in panel (**B**) represent 9 h red light/3 h white light/12 h dark and 9 h white light/3 h red light/12 h dark in a diurnal cycle. R:L = 12:0 (blue vertical dashed lines in panel (**A**) and L:R = 12:0 (blue vertical dashed lines in panel (**B**) represent 12 h red light/12 h dark and 12 h white light/12 h dark in a diurnal cycle.

**Figure 10 biology-11-01479-f010:**
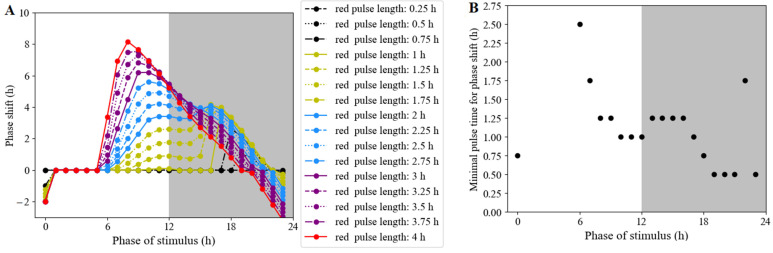
Phase-response curves measured under red-light pulses of various lengths. (**A**) The model was entrained under white-light/dark cycles (12 h light/12 h dark) for 5 days, then released to a red-light pulse at an indicated time. Each dot represents a pulse at an indicated time. (**B**) The minimum length of red-light pulse length required for producing an observable phase shift.

## Data Availability

Not applicable.

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
