# Peer review of "Red Light Resets the Expression Pattern, Phase, and Period of the Circadian Clock in Plants: A Computational Approach"

_biology, 2022, doi:10.3390/biology11101479_

Round 1

Reviewer 1 Report

I have some comments for the author to consider.

1. The manuscript title may need to be re-considered, I consider it should be specific.

2. Does this experiment not require plants for verification? What plants do you use?

3. How many repetitions are used for data in figures? How about the standard deviation?

4. The language should be polished.

5. The legend should be explained in the title of figures.

6. How to validate the model?

7. What are the applicability or use conditions of the model?

8. Please modify the format according to the guidelines.

9. What is the light intensity of red light and white light used in this experiment? The spectral composition of white/red light? Please provide more details in the Materials and Methods.

Author Response

Dear Reviewer,

Best wish!

Reviewer 2 Report

The manuscript by Huang et al., presents a model to incorporate light quality as an effector in models for plant circadian rhythm. The study is well conducted and will be of interest to the general as well as specialized audience. I recommend this manuscript for publication with some minor changes:

1. The authors developed a model for circadian rhythm in red-light/dark condition. The rationale for choosing red light as opposed to any other light condition is not very clear. Also, in natural settings plants seldom encounter light of a single wavelength. However the percentage of the wavelengths (i.e. colors) may vary depending upon the time of the day. The authors need to address this in the text and discuss how they can potentially use their model to simulate such conditions.

2. There are grammatical mistakes and missing words in several cases (including but not limited to lines 45, 96, 196).

3. The authors need to mention the full form of abbreviations (such as ODE), the first time they are mentioned.

Author Response

Dear Reviewer,

Best wish!

Reviewer 3 Report

This modeling article simulated the red- light entrainment of plant clock. However, they did not suggest a new biological mechanism of the entrainment. This paper is mostly focused on the mathematical calculation and proof that the calculation is fit to the experimental data which is published previously. I think this paper is more fit to the specific journal publishing the computational simulation research. If authors revise the paper by adding new mechanistic findings or applications of the simulation to explore the circadian clock mechanism, that may be considered biologically meaningful research.

Author Response

Dear Reviewer,

Best wish!

Round 2

Reviewer 1 Report

the manuscript can be accept

Reviewer 3 Report

The revised manuscript looks good for publication.